# Facile One-Step Synthesis and Enhanced Optical Nonlinearity of Graphene-γMnS

**DOI:** 10.3390/nano9121654

**Published:** 2019-11-21

**Authors:** Zhihao Zhang, Peng Li, Pengchao Li, Yuzong Gu

**Affiliations:** Institute of Micro/Nano Photonic Materials and Applications, School of Physics and Electronics, Henan University, Kaifeng 475004, China; kfzzh@163.com (Z.Z.); lilipengpeng@vip.henu.edu.cn (P.L.); pengchaoli158566@163.com (P.L.)

**Keywords:** graphene, graphene-γMnS, Z-scan, NLO property enhancement

## Abstract

Graphene-γMnS were prepared by facile one-step hydrothermal method. Structures and properties of samples were explored by characterization, and nonlinear optical (NLO) enhancement of nanocomposites (NCs) was fully studied. Nanoparticles and NCs were tested at 532 nm by a Z-scan technique. With γMnS attached in G layers, NLO susceptibility of graphene-γMnS was greatly improved under single-pulse laser irradiation compared with G and γMnS. The nonlinearity enhanced was attributed to local field effect and charge transfer between γMnS and graphene layers. And NLO property enhancement was restricted by non-radiative defects in graphene-γMnS. Exploring the mechanism of nonlinearity enhancement was significant for fabrication of NLO devices. However, the optical nonlinearity decreased first and then increased with further increased addition of GO, because the dispersion of γMnS attached on graphene surface might make density of sp^2^ fragment and defects changed. Graphene-γMnS exhibited excellent and tunable NLO performance, illustrating that NCs materials have potential applications in NLO devices.

## 1. Introduction

Recently, research on graphene and graphene-semiconductor are rapidly developing research fields [1]. Graphene has attracted extensive attention in basic research and application due to its two-dimensional nanostructure [2]. It is a kind of based material which has potential applications for various kinds of nonlinear optics devices [3]. Graphene exhibits important nonlinear optical (NLO) property because of its low band gap and high transparency [4].

Single graphene has weak NLO absorption and refraction, which can not meet the demand for various optical devices [5]. However, recent studies have reported that various kinds of graphene-semiconductor have excellent NLO property providing potential applications in NLO devices [6,7]. NLO response of nanocomposites (NCs) is many times larger than single graphene and nanoparticles, and different mechanisms have been proposed. Graphene-CuO exhibited significantly improved NLO response, because doping of CuO increased time of electron transfer and photon transitions, which inhibited the recombination between electrons and holes [8]. NLO susceptibility of graphene-CdFe_2_O_4_ enhanced was due to extent of conjugation with G layers [9]. In case of graphene-Pt, an enhanced NLO property was observed because of not only combination of light-induced electrons and energy transfer, but also the way their combined [10]. NLO property enhanced of graphene-TiO_2_ could be attributed to the combination of NLO absorption mechanism and NLO scattering [11]. γMnS is widely used in various fields as a common semiconductor material [12]. Like NLO property of graphene-CuO, graphene-CdFe_2_O_4_, graphene-Pt, and enhanced graphene-TiO_2_, electrochemical property of graphene-γMnS was also many times higher than that of γMnS nanocrystal and graphene [12,13,14]. However, there are few reports about NLO performance of graphene-γMnS. In fact, we have previously studied αMnS/rGO without thoroughly investigating the nonlinearity of graphene-γMnS and the mechanism of NLO performance enhanced of NCs [15]. It is necessary to explore NLO responses of graphene-γMnS and its mechanism of enhanced nonlinearity. 

In this study, graphene-γMnS was synthesized by facile one-step hydrothermal method, and mechanism of NLO enhancement was discussed. Nonlinearity of NCs was controlled by changing amount of GO added, and samples were tested at 532 nm by picosecond (PS) laser pulse. We want to explore whether graphene-γMnS had potential applications in optical communication, optical limiter, and all-optical switch.

## 2. Experiments

### 2.1. Synthesis of Graphene Oxide ( GO) and Graphene-γMnS

GO was prepared by improved Hummer method [16]. Firstly, graphite powder was oxidized to GO by KMnO_4_ and 98% H_2_SO_4_. Then, a solution was mixed with H_2_O_2_ and the mixture was washed several times with deionized water to remove impurities. Finally, products were dried at 45 °C for 48 h in vacuum drying oven. Improved Hammer method increased generation efficiency and oxidation degree of GO. Furthermore, this method could effectively prevent generation of toxic gases which caused harm to human body. High efficiency, high quality, and no toxicity in GO synthesis were important for large-scale production of GO.

Graphene-γMnS was prepared by facile one-step hydrothermal method. The synthesis process of NCs was shown in Figure 1. Firstly, GO was dispersed in ethylene glycol. Secondly, TAA and MnCl_2_·4H_2_O were added to GO suspension. After half-an-hour stirring, the solution was transferred to Teflon-lined stainless-steel autoclave and reacted at 170 °C for 6 h. The reaction temperature in our previous study was 190 °C and product was different [15]. Then, products were washed by anhydrous ethanol and deionized water. Finally, samples were dried in vacuum dryer at 45 °C for 48 h and labeled as Sample 1 (S1). Five samples are labeled as Sample 1 (S1), Sample 2 (S2), Sample 3 (S3), Sample 4 (S4), and Sample 5 (S5), which are obtained same experimental steps with different addition of GO. S1 was added in 15 mg GO, S2 was added in 30 mg GO, S3 was added in 60 mg GO, S1 was added in 90 mg GO, and S1 was added in 120 mg GO.

### 2.2. Instrumental Characterization

Graphene-γMnS were tested by X-ray diffraction (XRD, Bruker D8 Advance, Bruker Inc., Karlsruhe, Badensko-Wuertembersko, Germany), scanning electron microscope (SEM, Carl Zeiss Inc., Oberkochen, Baden-Württemberg, Germany), and transmission electron microscopy (TEM, JEOL JEM-2100 operating at 200 kV, JEOL Ltd. Inc., Akishima, Tokyo, Japan). FRIR spectra and Raman spectra were obtained on Bruker Optics Vertex 70 (Bruker Inc., Karlsruhe, Badensko-Wuertembersko, Germany) and Renishaw inVia (Renishaw Inc., Gloucester, Gloucestershire, UK), respectively. Ultraviolet Visible absorption spectra were acquired on Ultraviolet Visible absorption instrument (Uv-Vis, Cary 5000, Agilent Inc., Sacramento, CA, USA). The Z-scan patterns were received on picosecond laser (picosecond laser, PLA2251A, Ekspla Inc., Vilnius, Lithuania) with a wavelength 532 nm and the pulse width 30 ps. 

## 3. Results and Discussion

### 3.1. Structure and Morphology Characterization

XRD patterns of graphene, GO, graphene-γMnS and γMnS were shown in Figure 2. For GO and graphene, two peaks of 11° and 22° corresponded to crystal planes (001) and (002), illustrating that the bonding of oxygen and carbon atoms formed oxygen-containing functional groups, which were introduced in graphene [17]. High temperature and pressure removed oxygen-containing functional groups and made large sp^2^ domains formed, indicating that γMnS could be attached in graphene [18]. For pure γMnS, diffraction peaks of impurities could not be detected and the characteristic diffraction peaks of NCs were located at 26°, 28°, 29°, 46°, 50°, and 53° corresponding to (100), (002), (101), (110), (103) and (112), which illustrated that products were pure γMnS with wurtzite-type structure [19]. And characteristic peaks of NCs corresponded to nanoparticles which indicated that γMnS nanoparticles grew successfully in graphene layers. The XRD patterns of αMnS and αMnS composites in our previous studies were shown that characteristic diffraction peaks of composites were located at 29.6°, 34.3°, 49.3°, 59.3°, 58.5°, 61.4°, and 72.3° corresponding to (111), (200), (220), (311), (222), and (400) [15]. Figure 2 also showed that crystallinity of γMnS reduced as the amount of graphene increased. The larger amount of graphene added, larger chance for γMnS nanocrystals attached on surface of graphene. This resulted in crystallinity of γMnS decreased.

The morphology of GO and NCs were exposed to SEM in Figure 3. Figure 3a displayed that the surface of GO had many folds. The γMnS nanocrystals exhibited a granular structure. In S1 sample, since the amount of GO added is small, there were fewer places where γMnS could be attached, and γMnS were stacked. As the amount of GO added was continuously increased, the degree of dispersion on the surface of graphene decreased first and then constant. This was consistent with the reduction in crystallinity of the nanocrystals shown in the XRD pattern. Nanoparticles attached on graphene also made the smooth surface of grapheme become rough and surface defects increase. In order to get more information of NCs, EDX spectra of graphene-γMnS was recorded in Figure 3g. The weight percentages (weight%) of all elements were clearly described in the spectra, 29.98% for Mn, 11.12% for S, and 58.90% for C, indicating that γMnS nanoparticles were attached in graphene.

Further information about structure could be seen in TEM. Figure 4a displayed that the size of γMnS was about 100 nm. It could be obviously observed from Figure 4b that graphene was multi-layered with granular γMnS nanocrystals attached in. GO was reduced by high temperature, and γMnS nanocrystals were easier to adhere and grow on graphene surface. C=O, C-OH, and other functional groups on GO provided reactive anchoring sites for nucleation forming and growth of γMnS. 

More information about graphene-γMnS, γMnS, graphene, and GO could be obtained by FTIR spectra. Figure 5 showed FTIR spectra of all samples. It could be seen in Figure 5 that characteristic peaks at 3411 cm^−1^, 1632 cm^−1^, 1571 cm^−1^, 1239 cm^−1^, 1065 cm^−1^ and 627 cm^−1^ corresponded to stretching vibration of O-H, C=O, C-C, C-OH, C-S, and Mn-S, respectively [19,20,21]. When graphite particles were oxidized to GO, these oxygen-containing functional groups were formed exhibiting characteristic peaks at 3411 cm^−1^, 1632 cm^−1^, 1571 cm^−1^ and 1239 cm^−1^. When graphene-γMnS were formed, characteristic peaks at 1571 cm^−1^, 1065 cm^−1^ and 627 cm^−1^ were exhibited. At the same time, the characteristic peaks at 3411 cm^−1^ weakened, and 1632 cm^−1^ and 1239 cm^−1^ disappeared, demonstrating that large sp^2^ domains had been formed and smaller sp^2^ fragments existed between larger sp^2^ domains [22,23]. 

The UV-is absoption spectra of GO, γMnS and graphene-γMnS could be observed in Figure 6. The line of GO exhibited absorption peak at 235 nm, because π-π* transport at sp^2^ site [24]. And γMnS displayed a strong exciton absorption at 272 nm [25]. Figure 6 showed that a redshift of 8 nm could be seen in the line of graphene-γMnS compared with that of γMnS, which demonstrated that covalent attachment between γMnS and graphene sheets with some variation of electronic state of γMnS [3]. There was possible electronic transmittance between γMnS and graphene.

### 3.2. NLO Property of NCs

In our previous study, the NLO response of *α*-MnS/rGO was investigated, and results showed that the nonlinearity of NCs was enhanced, but the mechanism of enhancement was unclear. In this study, NLO absorption and refraction of samples could be measured by Z-scan technique using a single Gaussian beam. The transmittance T was measured as a function of laser incident energy density. The Nd:YAG laser system used for excitation was 30 PS laser pulse at 532 nm producing a repetition rate of 10 Hz, and the beam waist radius was about 10.6 μm at the focal plane. CS_2_ is used to calibrate the Z-scan curve so that the center of the curve is at the center of the *x*-axis. Nonlinear material CS_2_ was used to calibrate Z-scan and measured data could ignored the samples’ absorption and scattering effects. A cuvette was mounted on mobile platform controlled by computer, which moved samples along the focal plane of Z-axis and 250 mm focal length lens. Absolute ethanol was as a solvent and samples’ concentration was 0.125 mg/mL. The input single pulse intensity of the focal plane was adjusted to 15.1 GW·cm^−2^.

It could be seen in Figure 7a that error bars were in OA Z-scan curves of graphene. The difference may be due to the error of the laser’s source and signal receiver. Figure 7b,c presented open aperture (OA) and close aperture (CA)/OA Z-scan curve of graphene-γMnS, graphene, and γMnS. The curve of graphene exhibited a symmetrical peak, implying saturable absorption (SA). And the OA Z-scan curve of γMnS exhibited a symmetrical valley, implying two-photon absorption (TPA). However, the OA Z-scan curve of graphene-γMnS showed that a valley appeared in the peak at the focus, demonstrating that two-photon absorption appeared following SA. Figure 8b displayed that nonlinear refractive index of graphene, nanoparticles, and NCs is positive corresponding to the self-focusing effect, and NLO refraction of NCs was enhanced. 

OA Z-scan transmittance T could be calculated as [26]
(1)$$T(z)=\sum_{m=0}^{\infty }\left\{{\left[q_{0}\left(z\right)\right]}^{m}/{\left(1\;+\;m\right)}^{3/2}\right\},$$
where q_0_(z) was obtained by q_0_(z) = βI_0_L_eff_/(1 + z^2^/z_0_^2^) [26]. β was a nonlinear absorption coefficient which could be calculated as

(2)$$\beta \;=\;\left[2\sqrt{2}\left({1\;-\;T}_{z=0}\right)\left({1\;+\;Z}^{2}{\;+\;Z}_{0}^{2}\right)\right]{/(I}_{0}L_{\mathrm{eff}}).$$

L_eff_ was effect length which could be obtained by

L_eff_ = (1 − exp(−αL))/(αL).(3)

Imaginary part (Imχ^(3)^) and real part (Reχ^(3)^) could be obtained by Imχ^(3)^ = cn_0_λβ/480π and Reχ^(3)^ = n_0_n_2_/3π, where n_2_ was a nonlinear refractive index calculated as

n_2_ = (2.941 × 10^6^λω_0_n_0_τ△T_p-v_)/[EL_eff_(1 − S)^0.25^].(4)

So nonlinear susceptibilities of γMnS and NCs could be obtained as [27]

(5)|χ(3)| = [(Reχ(3))2 + (Imχ(3))2]1/2.

The susceptibilities of graphene, γMnS, and graphene-γMnS sample could be calculated by the above formula. It could be seen in Table 1 that NLO property of graphene-γMnS were obviously enhanced. χ^(3)^ of γMnS was 1.21 × 10^−12^ esu, χ^(3)^ of graphene was 0.78 × 10^−12^ esu, and χ^(3)^ of graphene-γMnS was 6.23 × 10^−12^ esu. These demonstrated that χ^(3)^ of graphene-γMnS was about five times larger than that of pure nanoparticles and about eight times larger than that of graphene. 

The data in Table 1 above showed that optical nonlinearity of graphene-MnS was better than combination of two separate components at high incident intensity. In an external electric field, the light-induced effective local electric field redistribution created an additional synergistic effect on NCs surface between γMnS and graphene layers, which could theoretically address the local field effect through the Maxwell Garnet model [28,29,30,31]. The model assumed that spherical particles, which had a diameter smaller than the wavelength of incident light, were encapsulated in a continuous host medium. Actual value of NLO susceptibilities were given in Table 1. The theoretical value of χ^(3)^ could be calculated and it was much larger than the sum of χ^(3)^ of two separate components. This was consistent with the measured results, indicating that local field effect was one of the significant reasons for NLO property enhanced of NCs materials [32]. Charge transfer (CT) has an important influence on the nonlinearity of NCs. CT from γMnS to graphene might produce additional synergistic effects of nonlinear enhancement [33]. Figure 8a showed that CT could be observed by changing incident intensity. The OA Z-scan curves of graphene-γMnS at different incident intensities were shown in Figure 8a. At low intensity such as 6.6 GW·cm^−2^, NCs exhibited SA. As the intensity increased to 8.5 GW·cm^−2^, 9.4 GW·cm^−2^, 15.1 GW·cm^−2^, 17.9 GW·cm^−2^, 18.9 GW·cm^−2^ and 20.8 GW·cm^−2^, a valley appeared in the peak at the focus and was getting deeper with incident intensity increased, which meaned that there was CT between them in two components [34]. The peak-to-valley development of OA Z-scan curves at different input intensities could further demonstrate the CT evolution in graphene-γMnS. The peak-to-valley development of OA Z-scan curves at different input intensities could further demonstrate the CT evolution in G-γMnS. *T_p_* and *T_v_* were normalized values of peak-to-1 and 1-to-valley of OA Z-scan curves, respectively, and *T_p-v_ = T_p_ + T_v_* [33]. It could be seen in Figure 8a that value of *T_p-v_* increased from 9.4 GW·cm^−2^ to 20.8 GW·cm^−2^, indicating that CT through interface was in this intensity range. And CT could not be observed at intensity below this range. To further prove CT only could be obviously observed at high incident intensity, χ^(3)^ of samples at 6.6 GW·cm^−2^ were shown in Table 1. NLO susceptibility of NCs was 4.85 × 10^−12^ esu at low incident intensity. Larger χ^(3)^ of graphene-γMnS demonstrated that CT appeared between graphene and graphene-γMnS at high incident intensity and no obvious CT at low incident intensity. Fluorescence experiments could further prove that there was CT between donor-receptors. Figure 8b showed fluorescence spectra of γMnS, graphene, and graphene-γMnS. The excitation wavelength was 283 nm and fluorescence peak of γMnS appeared at 432 nm. However, graphene-γMnS did not show fluorescence peaks, demonstrating that the trend of synergistic electron transfer was between two components [4]. The NLO susceptibility of graphene-γMnS was increased by an order of magnitude, which might be due to defects. In CT process, since defect sites provided a position for their conjugation and capture excited electrons. Radiation defects were effective for electron transport, which made trapped electronscould be released to low-energy states. 

The information of NCs surface structure could be obtained by Raman spectroscopy. The density of NCs surface defects could be obtained by I_D_/I_G_ ratio. Surface defects had an important influence on the nonlinearity of NCs. Figure 9a showed Raman spectra of S1, GO, and graphene, which exhibited D and G bands at 1346 cm^−1^ and 1583 cm^−1^, respectively [22]. The I_D_/I_G_ ratio of NC was obviously larger than that of gaphene, indicating that NCs had more surface defects and nanoparticles were attached in G. For all NCs in Figure 9b, I_D_/I_G_ ratio increased first and then decreased as the amount of graphene increased. The increase of the ratio indicated that there were fewer defects due to the densities of γMnS decreased on the surface of graphene, while the decrease of the ratio represented that NCs with little GO addition had more defects. This was consistent with the results of SEM and XRD. For graphene-γMnS, the average sp^2^ domain size or the effective in-plane correlation length of the sp^2^ domain size *L_a_* were given by this formula $L_{a}\;=\;1.8\;\times \;10^{-9}\;\lambda_{plaser}^{4}{/(I}_{D}{/I}_{G}$), where *λ_plaser_* was the wavelength of the test laser (532 nm) [24]. The defect density *n_D_* could be approximately calculated by formula $n_{D}\;=\;1.8{\;\times \;10^{22}\;(I}_{D}{/I}_{G})/\lambda_{plaser}^{4}$ [25]. The results were shown in the Figure 9c, indicating that sp^2^ fragment density, which decreased first and then increased. Defects affected the local state of a small area sp^2^ domain and the large area sp^2^ cluster on graphene.

Figure 10a,b showed OA Z-scan curves and OA/CA Z-scan curves of S1, S2, S3, S4 and S5. The nonlinear optical parameters of S1, S2, S3, S4 and S5 was obtained from Table 2. Because the band gap of GO reduced was zero, when incident light was irradiated onto the NCs surface, a large transient carrier group was generated in the conduction band (CB) and the valence band (VB). Since the pulse duration was comparable to the carrier-to-band relaxation time, when a strong laser pulse was applied, more electron-hole pairs were generated and resulted in CB state filling and VB bleaching, which would prevent further absorption and lead to SA behavior [35]. Due to the relatively large density of states in the metal, the probability of the excited electrons in the CB of graphene shifting to the metal had higher probability than that of shifting to VB of graphene. Because the carriers were excited faster than their relaxation from the γMnS nanoparticles to the VB of graphene, the bleaching of the ground state occured, resulting in nonlinearity enhanced [36]. The trend of nonlinearity was decreased first and then increased, which might be related to the change of local state on graphene surface.

The transfer from the sp^3^ matrix to the sp^2^ domain and the sp^2^ cluster directly affected the CT between the graphene and the γMnS nanoparticles, while the sp^3^ matrix played an important role on the active site for the attachment of the γMnS nanoparticles. Because the local state of the small area sp^2^ domain and the large area sp^2^ cluster were related to the CT time and relaxation time, which influenced changed nonlinearity. More local states on graphene surface increased CT time, resulting in the reduction in nonlinearity of NCs. As the addition of GO increased, the nonlinearity of NCs decreases first. It could be seen from SEM, XRD and Raman patterns that the distribution state of γMnS on graphene surface was greatly affected by the amount of GO added. For S1, S2, and S3, γMnS nanoparticles were stacked on the GO surface with a low degree of dispersion, which resulted in a large number of smaller sp^2^ domains growing. These lead to percolation between the larger sp^2^ clusters via growth of smaller sp^2^ domains, which increased the spatial overlap of electron and hole wave functions, thereby reducing the oscillator strength of excitons [37]. As amount of GO added continued to increase, the nonlinearity of NCs began to increase. γMnS nanoparticles began to disperse on graphene surface because more attachment points could make nanoparticlse grow, which also reduced the sp^2^ domain of the small segment and did not interconnect the larger sp^2^ clusters, which led to increase in electron-hole pairs. The overlap of electron and hole wave functions reduced, resulting in the nonlinearity enhanced of NCs.

## 4. Conclusions

In summary, γMnS and graphene-γMnS composites were synthesized by hydrothermal method. The structure, absorption and refraction of γMnS and graphene-γMnS composites were characterized. The NLO property were investigated by 532 nm pulsed laser with pulse duration of 30 ps. The NLO enhancement of NCs was due to included local field effects and charge transfer. However, results of theoretical calculations were deviated from experimental data, and NLO enhancement was within an order of magnitude due to non-radiative defects in graphene-γMnS. This article also provided a facile approach of changing addition of GO to obtain tunable NLO property of NCs. As the addition of GO increased, NLO characteristics of NCs decreased first and then increased. Structure of NCs made local state of the small area sp^2^ domain, the large area sp^2^ cluster on graphene, and defects changed. Tunable NLO performance of NCs by adjusting defects was important for the fabrication of nonlinear optics such as optical switches, optical sensors, and so forth.

## Figures and Tables

**Figure 1 nanomaterials-09-01654-f001:**
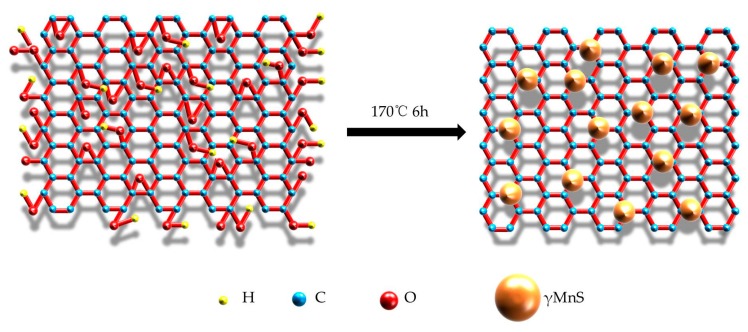
The process of γMnS attached in GO.

**Figure 2 nanomaterials-09-01654-f002:**
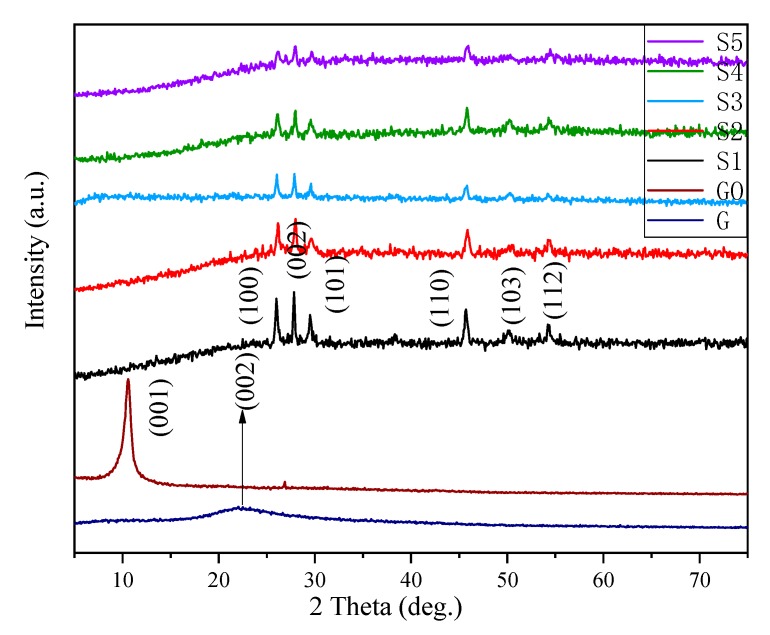
XRD patterns of graphene, GO, NCs, and γMnS.

**Figure 3 nanomaterials-09-01654-f003:**
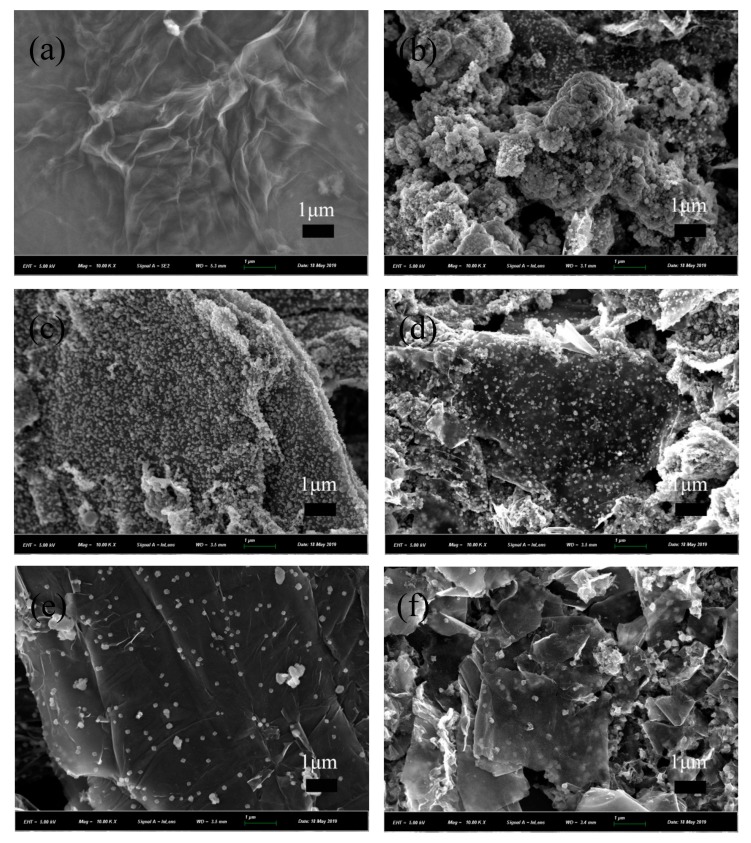
SEM images of (**a**) GO, (**b**) S1, (**c**) S2, (**d**) S3, (**e**) S4, and (**f**) S5. (**g**) EDS spectra of graphene-γMnS.

**Figure 4 nanomaterials-09-01654-f004:**
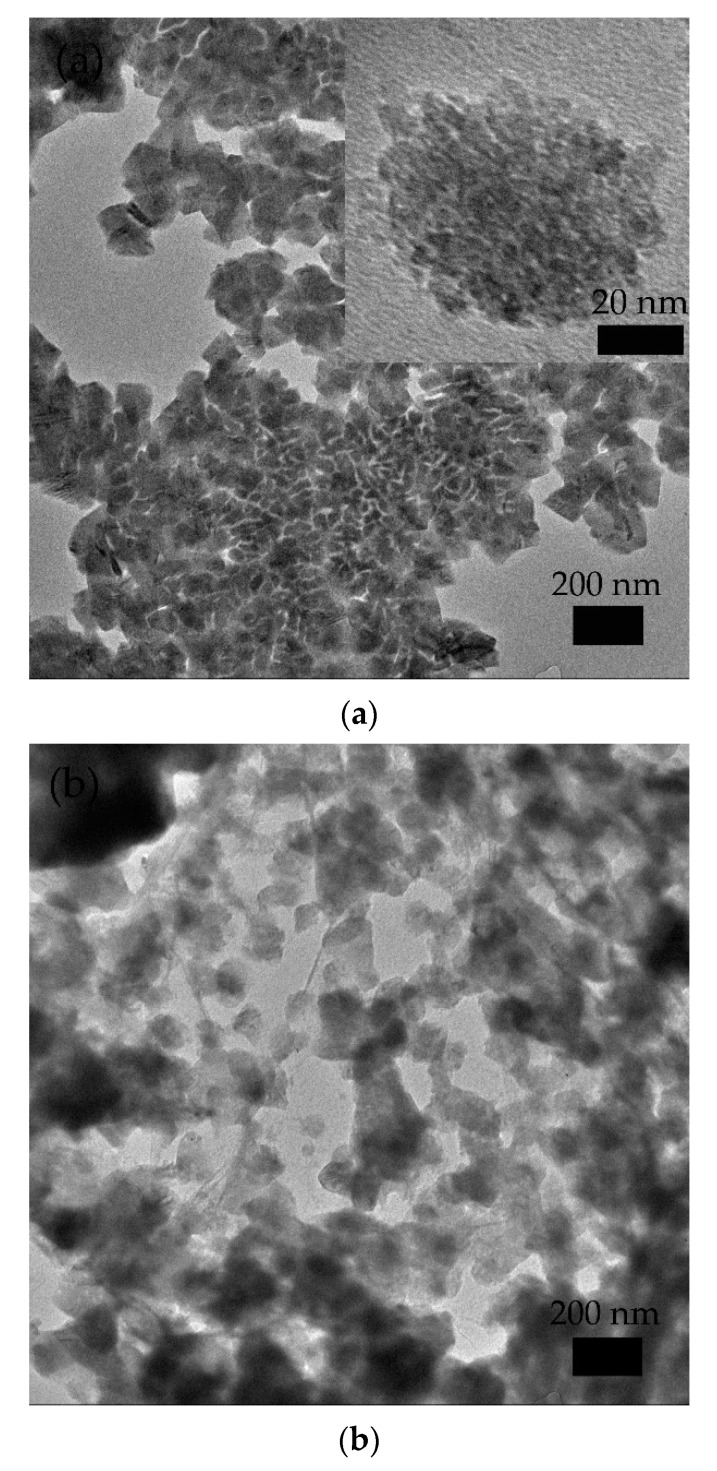
TEM images of (**a**) γMnS and (**b**) graphene-γMnS.

**Figure 5 nanomaterials-09-01654-f005:**
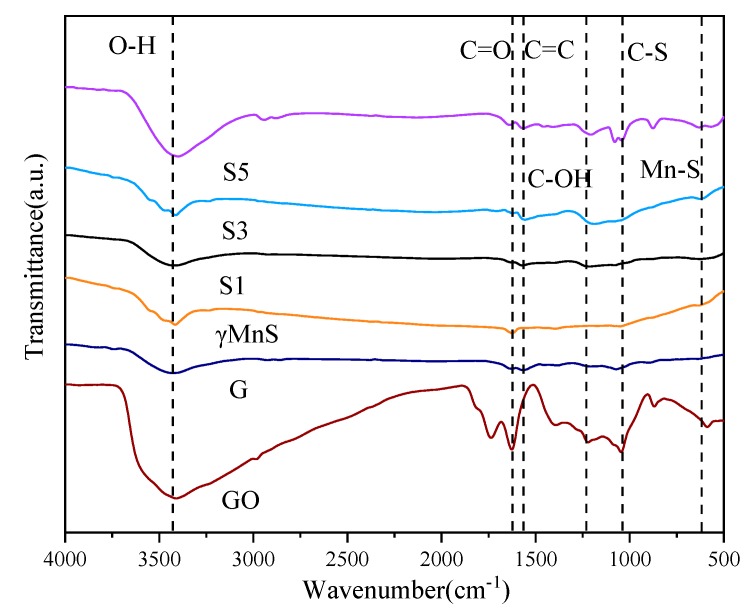
FTIR spectra of NCs, γMnS, graphene and GO.

**Figure 6 nanomaterials-09-01654-f006:**
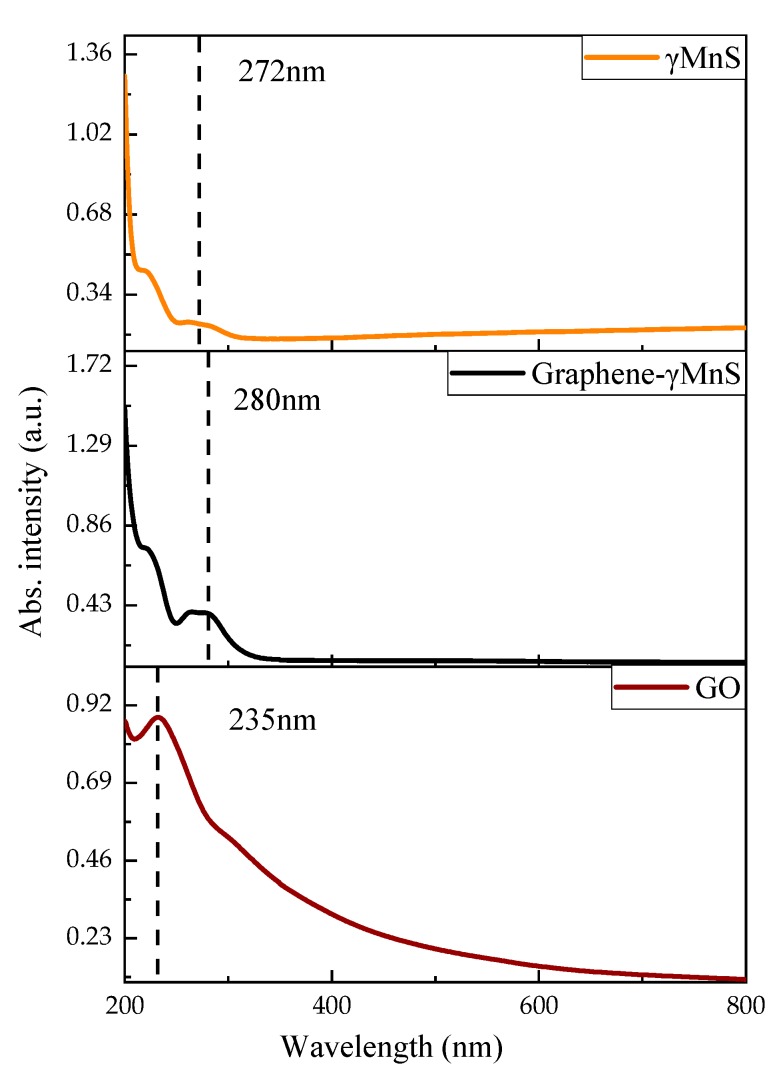
UV-vis absorption spectra of GO, γMnS, and graphene-γMnS.

**Figure 7 nanomaterials-09-01654-f007:**
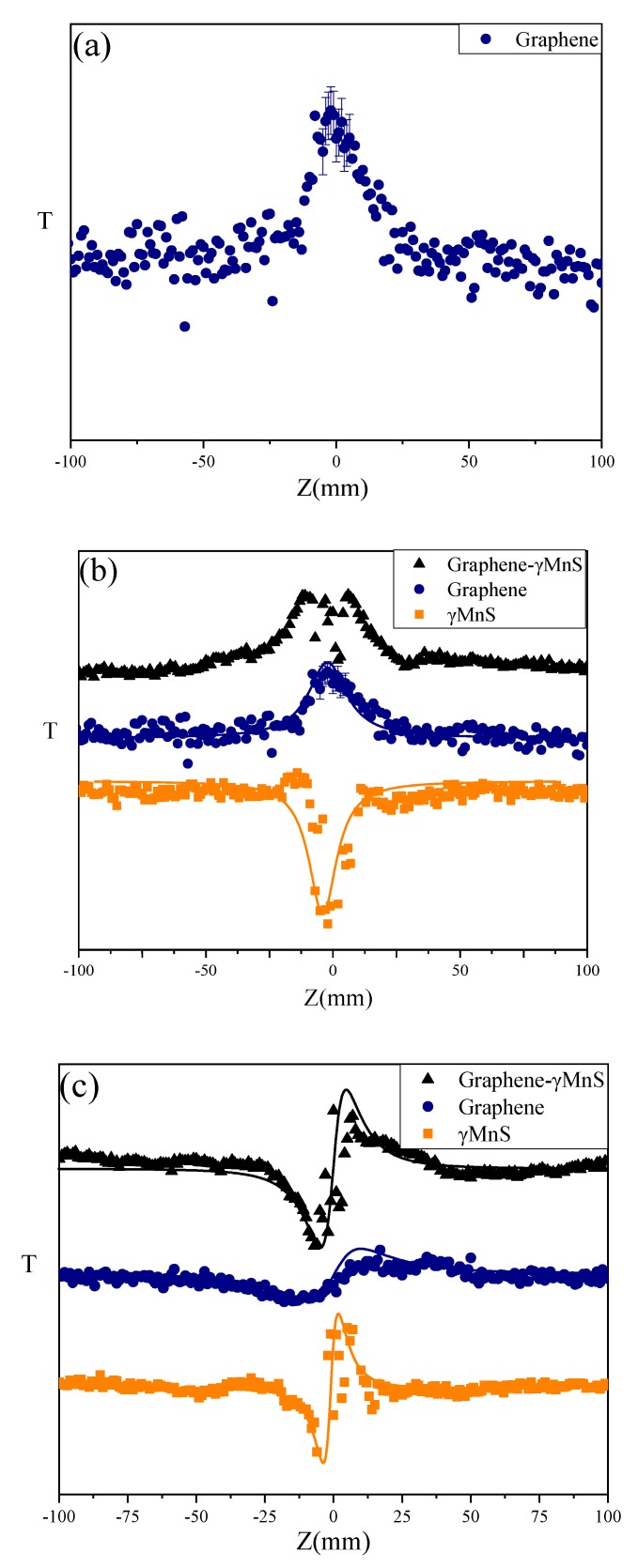
(**a**) OA Z-scan curves of graphene (**b**) OA Z-scan curves of graphene-γMnS, graphene, and γMnS at 15.1 GW·cm^−2^. (**c**) CA/OA Z-scan curves of graphene-γMnS, graphene, and γMnS at 15.1 GW·cm^−2^.

**Figure 8 nanomaterials-09-01654-f008:**
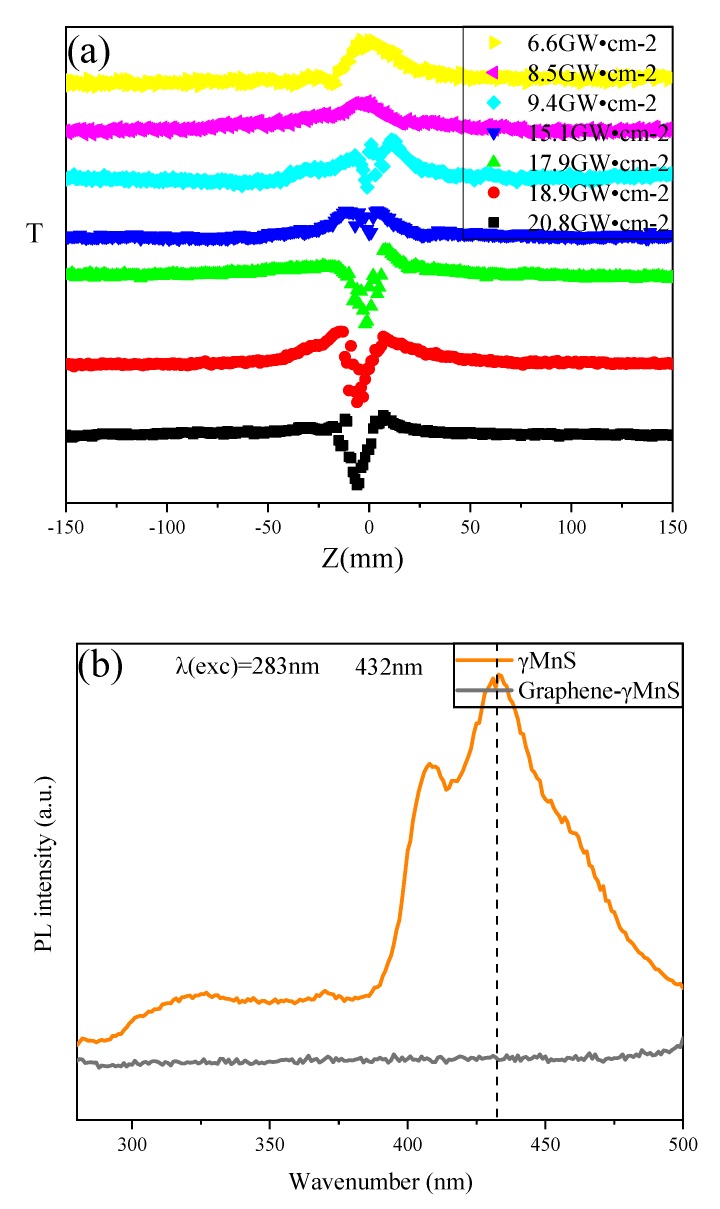
(**a**) OA Z-scan curves of graphene-γMnS at 6.6 GW cm^−2^, 8.5 GW cm^−2^, 9.4 GW cm^−2^, 15.1 GW cm^−2^, 17.9 GW cm^−2^, 18.9 GW cm^−2^ and 20.8 GW cm^−2^. (**b**) PL patterns of graphene-γMnS and γMnS.

**Figure 9 nanomaterials-09-01654-f009:**
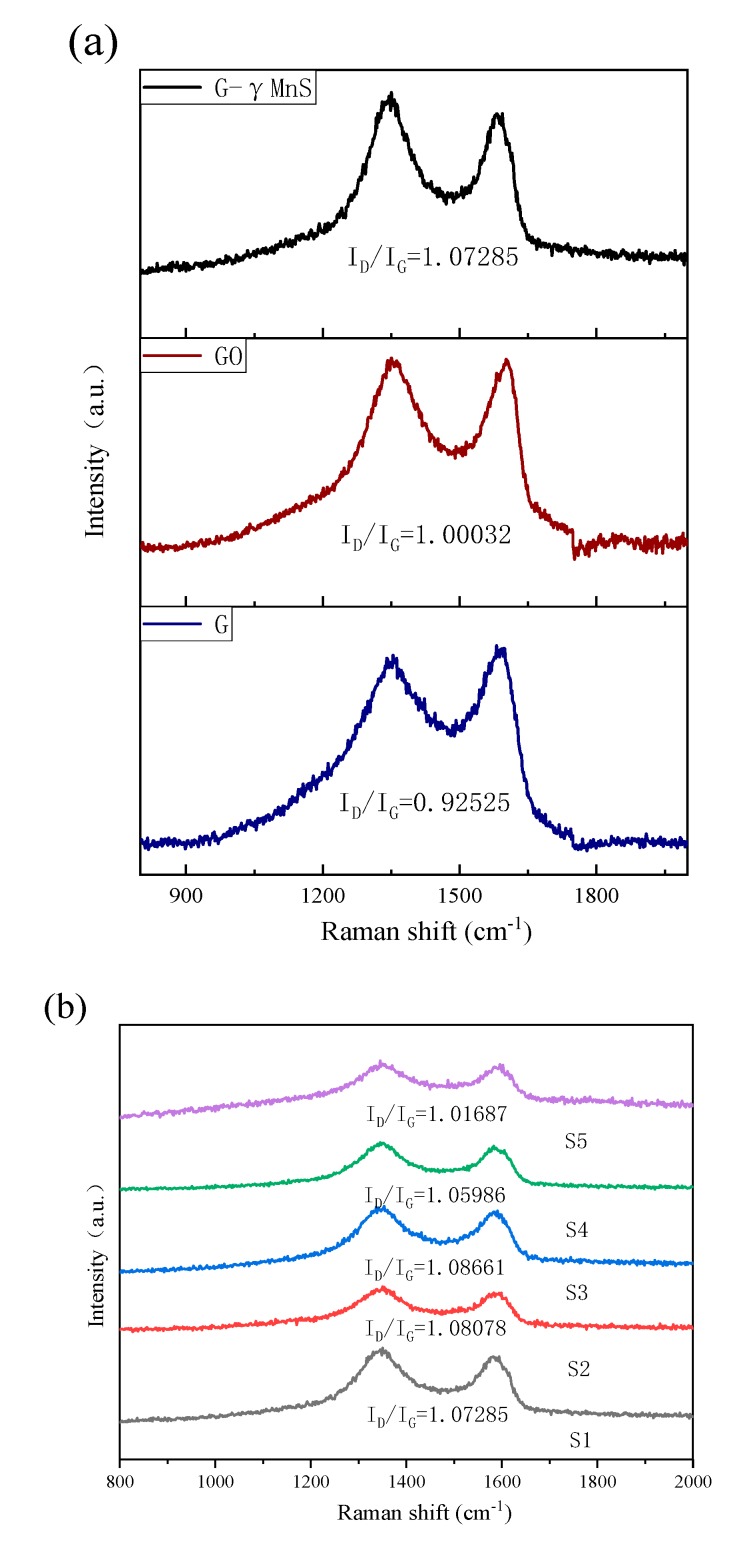
Raman spectra of (**a**) S1, GO and G. (**b**) S1, S2, S3, S4 and S5. (**c**) Defect density of all samples.

**Figure 10 nanomaterials-09-01654-f010:**
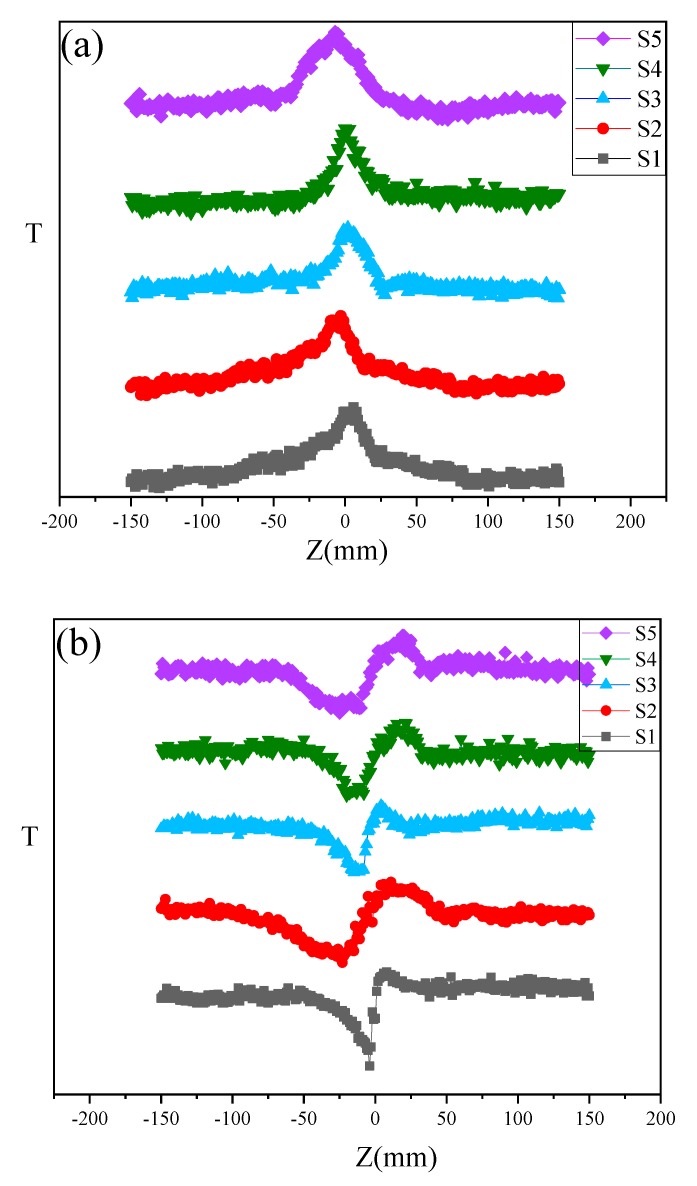
(**a**) OA Z-scan curves and (**b**) CA/OA of Z-scan curves S1, S2, S3, S4 and S5.

**Table 1 nanomaterials-09-01654-t001:** The nonlinear optical parameters of γMnS, graphene and graphene-γMnS.

Sample	Imχ^(3)^/10^−12^ esu	Reχ^(3)^/10^−12^ esu	χ^(3)^/10^−12^ esu
γMnS	0.96	0.74	1.21
graphene	0.65	0.42	0.78
graphene-γMnS (tested at 15.1 GW·cm^−2^)	3.44	5.19	6.23
graphene-γMnS (tested at 6.6 GW·cm^−2^)	4.78	1.56	5.03

**Table 2 nanomaterials-09-01654-t002:** The nonlinear optical parameters of S1, S2, S3, S4, S5.

Sample	Imχ^(3)^/10^−12^ esu	Reχ^(3)^/10^−12^ esu	χ^(3)^/10^−12^ esu	β/10^−11^ mW^−1^
γMnS	0.33	0.89	0.95	1.48
S1	0.97	4.78	4.88	2.31
S2	0.83	4.54	4.61	2.19
S3	0.67	4.42	4.48	2.13
S4	1.14	6.05	6.46	2.92
S5	2.73	6.86	7.39	3.28
G	0.46	1.32	1.4	0.64

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
