# Peer review of "Facile One-Step Synthesis and Enhanced Optical Nonlinearity of Graphene-γMnS"

_nanomaterials, 2019, doi:10.3390/nano9121654_

Round 1
Reviewer 1 Report
This version of the manuscript has satisfactorily considered my previously comments.
Author Response
Dear reviewer,
Thank you for your letter and for the comments concerning our manuscript entitled “Facile one-step synthesis and enhanced optical nonlinearity of graphene-γMnS”(ID: nanomaterials-638876). Thanks very much for the recognition of our article and the useful suggestions.

Reviewer 2 Report
Authors provide rather detailed investigation of the third-order nonlinear optical properties of gamma-MnS and graphene-gamma-MnS composites and discuss mechanisms for its enhancement. Authors exploit well known Z-scan techniques complemented with other supporting data obtained with traditional characterization methods (PL, Raman, x-ray, linear absorption). Authors have recently published similar study where other materials have been investigated in about the same experimental arrangements. I , overall , support acceptance of the manuscript for publication. However, the following issues have to be addressed.
How precisely the calibration with CS2 sample has been done ? Can they show the data (on an inset, for example). Can they select one Z-scan data curve (for graphene for example) and make a bigger plot as opposed to displaying 3 curves on one plot? Please display an error bar for the peak value on that Z-scan? Explain expected open aperture (OA) and closed aperture (CA) results and differences. Saturable absorption effect is claimed for graphene as authors observe 'symmetrical' nature of the Z-scan peak? How valid is this statement to make the conclusion? The same question on the presence of TPA as authors state that the 'symmetrical valley' suggest that. How difficult it is to perform the experiment with the fundamental frequency beam (i.e. at 1064 nm)? Will the two-photon absorption feature appear there as well? Expand explanations and provide more details for Z-acan data (for OA) at different power densities (i.e. Fig 8a) How do they exactly estimate the defect density for different samples presented in figure 9 (c) ? Error bars? The manuscript should be carefully read and edited by a person who has taken higher level English courses. Grammar and issues related to construction of sentences should be addressed in particular. Some of the sentences are not read well to say the least(!). e.g.
'Recently, graphene and graphene-semiconductor are rapidly developing research fields'
'The goal is to explore whether graphene-γMnS had potential applications in optical communication, optical limiter and all-optical switch.'
And many MORE within almost any paragraph.
Author Response
Dear reviewer,
Thank you for your letter and for the comments concerning our manuscript entitled “Facile one-step synthesis and enhanced optical nonlinearity properties of graphene-γMnS” (ID: nanomaterials-638876). Those comments are all valuable and very helpful for revising and improving our paper, as well as the important guiding significance to our researches. We have carefully studied the comments and have made correction which we hope to meet with approval. Revised portions are marked in red in the manuscript. The main corrections in the manuscript and the responds to the reviewer’s comments are as follow.
Point 1. How precisely the calibration with CS2 sample has been done? Can they show the data (on an inset, for example).
Response 1: CS2 is used to calibrate the Z-scan curve so that the center of the curve is at the center of the x-axis. The figure below is OA Z-scan curves of CS2.
Point 2. Can they select one Z-scan data curve (for graphene for example) and make a bigger plot as opposed to displaying 3 curves on one plot?
Response 2: Z-scan curve of graphene is shown below. But putting together three z-scan curves may make the difference between them more obvious.
Point 3. Please display an error bar for the peak value on that Z-scan? Explain expected open aperture (OA) and closed aperture (CA) results and differences.
Response 3: Error bars have been added in the NLO measurements as shown below. The difference may be due to the error of the laser's source and signal receiver.
Point 4. Saturable absorption effect is claimed for graphene as authors observe 'symmetrical' nature of the Z-scan peak? How valid is this statement to make the conclusion? The same question on the presence of TPA as authors state that the 'symmetrical valley' suggest that. How difficult it is to perform the experiment with the fundamental frequency beam (i.e. at 1064 nm)? Will the two-photon absorption feature appear there as well?
Response 4:
Saturated absorption means that when the intensity of the input light exceeds a threshold, this unique absorption property begins to become saturated. CB state filling and VB bleaching will prevent further absorption and lead to SA behavior. The OA curve is a peak indicating that graphene exhibits saturated absorption characteristics, rather than symmetrically indicating the NLO characteristics of graphene. The OA curve is a valley indicating that graphene composites exhibit TPA. Many studies in my group have shown that the scattering effect has no effect on the accuracy of the experimental data.
When the wavelength of the incident laser is 1064 nm, since the energy of the photon is small and does not provide enough energy, there is no absorption of two-photon phenomenon.
Point 5. Expand explanations and provide more details for Z-acan data (for OA) at different power densities (i.e. Fig 8a). How do they exactly estimate the defect density for different samples presented in figure 9 (c)? Error bars?
Response 5:
The peak-to-valley development of OA Z-scan curves at different input intensities could further demonstrate the CT evolution in G-γMnS. Tp and Tv were normalized values of peak-to-1 and 1-to-valley of OA Z-scan curves, respectively, and Tp-v = Tp + Tv [33]. It can be seen in Fig. 8 (a) that value of Tp-v increased from 9.4 GW·cm-2 to 20.8 GW·cm-2, indicating that CT through interface was in this intensity range. And CT could not be observed at intensity below this range.
The defect density of different samples can be estimated by the formula for calculating the defect density in the manuscript. This has been proved in the study of our group, and the following literature is the study of our group.
Point 6. The manuscript should be carefully read and edited by a person who has taken higher level English courses. Grammar and issues related to construction of sentences should be addressed in particular. Some of the sentences are not read well to say the least(!). e.g. 'Recently, graphene and graphene-semiconductor are rapidly developing research fields' 'The goal is to explore whether graphene-γMnS had potential applications in optical communication, optical limiter and all-optical switch.' And many more within almost any paragraph.
Response 6: Our manuscript has been revised. For example, ‘'Recently, research on graphene and graphene-semiconductor are rapidly developing fields’ 'We want to explore whether graphene-γMnS had potential applications in optical communication, optical limiter and all-optical switch.'

Round 2
Reviewer 2 Report
Authors have addressed some (but not all) points of criticism. The paper reads better now (but still can be improved) and presentation of the obtained results and discussions are more clear now.
Author Response
Dear reviewer,
Sorry to bother you because of our mistake and thank you for your letter and for the comments concerning our manuscript entitled “Facile one-step synthesis and enhanced optical nonlinearity properties of graphene-γMnS” (ID: nanomaterials-638876). Those comments are valuable and very helpful for revising and improving our manuscript. We have carefully studied the comments and have made correction which we hope to meet with approval.
We add Fig. 7 (a) and changed Fig. 8(a) and 9(c) as follow. And we have recorrected the language of the manuscript.

This manuscript is a resubmission of an earlier submission. The following is a list of the peer review reports and author responses from that submission.
Round 1
Reviewer 1 Report
1) Lines 150-160: Display all equations, and number them consecutively for the sake of clarity.
2) Table 2: Im{X(3)} cannot be negative otherwise it leads to nonlinear amplifications, which is physically impossible.
3) The widely used and recognized abbrevation for Graphene is Gr and not G. This has to be corrected throughout the text
4) Line 234: There is a reference to [36] which does not exist
Reviewer 2 Report
The manuscript described the synthesis of γMnS with the x-ray diffraction identification and its deposition on graphene oxides. Since the materials characterizations of resulting G-γMnS are standard techniques, therefore, the main focus will be on the discussion of nonlinear properties and interpretation of NLO data. There are certain needs required to improve the quality of manuscript, as follows.
(1) The title should be corrected as “Facile one-step synthesis and enhanced optical nonlinearity properties of graphene-γMnS”.
(2) Characterization on the surface bonding of γMnS to graphene oxide, after the treatment at 170 oC, was not clear. The assignment of C-S bonds at 1065 cm‒1 (page 5) is not sufficient to prove the formation of these bonds since it is in the same range of C-O absorptions. There is no reason to believe all oxides (C-O-C, C-O, C=O, etc.) on graphenes will be fully eliminated at 170 oC.
(3) If the NLO properties are proposed to arise from the charge-transfer (CT) transient states of G-γMnS (pages 8 and 9), the author should provide (1) nanosecond or picosecond transient absorption spectrum of G-γMnS to detect the corresponding absorptions of both (G)x‒ and (γMnS)x+ transient states and/or (2) cyclic voltammograms (CV) of G-γMnS to measure the reversible cyclic redox potentials of both G and γMnS components to prove the feasibility of the photoinduced CT states.
(4) Figure 9 is basically under the energy-transfer pathway. It is not necessary to provide the evidence of CT pathway. In fact, in the case of nanocomposites, both pathways are co-exist. This is the reason that nanosecond or picosecond transient absorption spectroscopic measurements are the necessary technique to give the direct evidence of photoinduced CT states.
Reviewer 3 Report
This manuscript could be of potential interest for Nanomaterials, but need some revisions.
-Part of the material related to the characterization of G-γMnS could be moved as supporting information.
This is particularly true for literature data related to Table 1.
For example, SEM or TEM images of G are very known from the literature, thus could be used for comparison just in an image of supporting information. Analogously, the UV spectrum of G is useless to be reported in the figure.
-Figure 6 should be corrected: what is S1 in that Figure? Moreover, how were recorded the Uv/vis absorption spectra? Are they recorded in transmission? In that case in ordinate of figure 6 should be reported the actual absorbance. In Figure 9 should be indicated the λ(exc).
-The discussion on page 9, lines 212-227 is unclear and for some aspects rather contradictory.
Round 2
Reviewer 3 Report
The revised manuscript is partly improved.